# Development of Ready-to-Eat Organic Protein Snack Bars: Assessment of Selected Changes of Physicochemical Quality Parameters and Antioxidant Activity Changes during Storage

**DOI:** 10.3390/foods11223631

**Published:** 2022-11-14

**Authors:** Aleksandra Szydłowska, Dorota Zielińska, Monika Trząskowska, Katarzyna Neffe-Skocińska, Anna Łepecka, Anna Okoń, Danuta Kołożyn-Krajewska

**Affiliations:** 1Department of Food Gastronomy and Food Hygiene, Institute of Human Nutrition Sciences, Warsaw University of Life Sciences (WULS), 02-776 Warsaw, Poland; 2Department of Meat and Fat Technology, Prof. Wacław Dąbrowski Institute of Agricultural and Food Biotechnology—State Research Institute, Rakowiecka St. 36, 02-532 Warsaw, Poland

**Keywords:** protein bars, whey protein concentrate, organic food, quality, storage

## Abstract

Novel organic high-protein bars (HPB) were developed and produced from organic ingredients such as prebiotic and pro-healthy additives or whey protein concentrate (WPC-80). The influence of temperature and time on the selected physicochemical parameters and antioxidant activity of three formulations of HPBs when stored (at 4 °C and 22 °C for 3 months) was investigated. The fresh products varied on the basis of available carbohydrates, crude lipids, amino acid profile, and fatty acid profile resulting from the used formulations. A total of 17 amino acids (AA), including 10 essential amino acids (EAA), were identified in HPBs. The concentrations of all essential amino acids determined by EAA scores (AAS), except Histidine (His), were higher than the FAO/WHO/UNU (2007) pattern; for the WPC-80 however, in the case of the developed HPB, the scores were lower (0.21–0.48). The first limiting amino acid in HPB was Val (Valine). The temperature and time of storage significantly affected the proximate chemical composition and an assessment of the products’ antioxidant activity. The amino acid and fatty acid composition of stored products slightly changed. However, stored HPBs had a low content of trans fatty acids (TFAs). The optimal method of storage for the investigated bars was at the temperature of 4 °C for 3 months.

## 1. Introduction

There is a current preference for health-related foods among consumers, including a low-calorie diet containing higher amounts of protein, fiber, and antioxidants, whilst also being easy to handle, store, and consume. The high value of ready-to-eat products, such as snack bars, can be attributed to their ease of use [1,2]. New opportunities for protein product development are offered by the growing food protein market, but they are also adding new challenges to food protein research and innovation [3,4]. Human growth and maintenance require energy and essential amino acids, both of which are found in food proteins. Furthermore, many food proteins exhibit specific biological activities that can influence human health and prevent diseases [3]. The quality of such food products is also important. Consumers are primarily concerned with the color and taste of food ingredients, as well as aspects of nutritional quality, including energy, vitamins, minerals, fiber, and many bioactive compounds that improve human health [5].

Consuming organic goods is another trend that has developed over the past ten years. Consumers associate organic food with favorable health effects because they believe it is dangerous to be exposed to man-made pesticides and fertilizers that are still present in traditional food [6]. Organic farming is generally based on natural fertilizers (such as animal and green manure); therefore, the total quantity of plant nutrients (specifically nitrogen) is lower in organic than conventional farming. Consequently, there is a reduction in the overall amount of nutrients that are accessible for plant development. However, some studies have shown that since the quality of the amino acids in organic products is greater than in conventional products, there are more essential amino acids present in organic grains [7]. Moreover, in most studies, higher levels of phenolic compounds, vitamins, and minerals were observed in organic plants in comparison with conventional ones [8].

The development of food quality depends on the course of various processes, both physicochemical and biochemical. According to Moschopoulou et al. (2019) [9], intrinsic factors, originating from the raw materials used, such as water activity, pH, and chemical and microbiological composition, and extrinsic factors, such as storage conditions and packaging, may affect the changes in the quality of food products during their shelf life. It is very rare for food to improve in quality with storage. Texture, flavor, color, and nutrition composition can all change, and their rates of change are often regulated by the conditions in which they are stored [10]. In the food production chain, storage conditions play an important role in product stability and quality. The industry requires proper methods of storing food to maintain quality and shelf life until the food is finally placed on the market [11].

In many countries, the consumption of snack bars is very popular and continuously increasing, which prompted the food industry to develop new cereal-bar formulations and ingredients [12]. Although, it is a challenge to develop a snack bar in manageable portions with high levels of protein, fiber, and antioxidants due to the need for the retention of nutritional compounds and antioxidant contents after processing and during storage. In this context, the overall objective of this study was to investigate the effect of temperature and storage period on selected physicochemical parameters (proximate chemical composition; amino acid composition; fatty acid profile) and antioxidant activity of organic protein bars to develop pro-health food for consumers. The novelty of the study includes the use of minimally processed organic ingredients to develop snack bars with high protein, fiber content, and antioxidant activity and assess the optimal storage conditions taking into account nutrient quality. The study is the continuation of previous research [13] aimed at searching for the best HPB formula.

## 2. Materials and Methods

### 2.1. Materials

The study material includes three series of protein bars, which were made according to the formulation shown in Table 1. Three batches of organic protein snack bars were studied and all analyses were performed in three independent replicates.

The basic ingredients were certified organic food purchased from a Polish manufacturer (Symbio Co., Lublin, Poland). All ingredients used in this study were purchased and kept at room temperature until further use. The organic WPC-80 (Symbio Co., Lublin, Poland) contained 80% protein according to the manufacturer’s declaration. All chemicals used were of analytical grade.

#### 2.1.1. Procurement of Used Ingredients

All ingredients were weighed in amounts according to Table 1. Pumpkin seeds, spelt flakes, oatmeal, and coconut shreds were crushed. Oatmeal was additionally roasted in a frying pan and cooled. The dried plums or/and dried apricots were poured with a boiling point and set to soak for 1 h and then water was drained and the fruits were mixed to a smooth weight. The chocolate coating was also prepared. The chocolate was dissolved in a water bath and then WPC-80 and the appropriate water portion (Table 1) were added. Proportionately, 10 g of WPC-80 was added for each 100 g of chocolate. At last, everything was mixed and slightly cooled.

#### 2.1.2. The Organic Protein Snack Bars Preparation and Storage Procedure

After combining all the prepared ingredients of the recipe, the homogeneous mass was placed in silicone molds and cooled (refrigeration conditions, at 4 °C for 4 h). The bars were chocolate coated and the products were cooled again until the chocolate mass solidified. Each bar of weight 40 g was cut into a rectangular piece and packed into metalized polyester polyethylene (MET-PPE) and sealed. The packaging process was conducted at ambient temperature 24 ± 2 °C. The products were stored at 4 °C for 10 h and then used for analysis.

The samples of HBPs were subjected to a controlled time of storage (for three months) and two temperature conditions (4 °C and 22 °C).

### 2.2. Methods

#### 2.2.1. Chemical Analyses

Physicochemical Composition Analysis and Determining Caloric Value

For proximate composition analysis, the formulated organic protein bars were crushed in mortar and pestle. The moisture, ash, crude fat, crude fiber, total protein (Kjeldahl method, N × 6.25), and total reducing sugars content were determined according to standard AOAC Official Method 934.06 (2006), AOAC Official Method 923.03 (2006), AOAC Official Method 920.39 (2006), AOAC Official Method 985.29 (2006), AOAC Official Method 945.18-B (2005), and AOAC Official Method 945.66 (1945) [14], respectively. The content of available carbohydrates was calculated by a difference from the total protein, lipids, moisture, ash, and crude fiber contents. The Atwater general factor system was used to calculate caloric values: carbohydrate (4 Kcal g^−1^), fat (9 Kcal g^−1^), and protein (4 Kcal g^−1^) (Regulation (EU) No 1169/2011) [15].

Amino acid composition

The AAs compositions in the WPC-80 and HBP were analyzed. The analysis of amino acids was based on the method of sample hydrolysis and the analysis of the individual amino acids using HPLC (High-Performance Liquid Chromatography) (Agillent 1100 Series HPLC, Waldbronn, Germany) with automated on-line Pre Column Derivatization (with the use a liquid sampler and Poroshell 120 column HPH-C18 (3 × 100 mm, 2.7 μm. P/N 695975–502). The total amino acid content was measured using two methods, including acid hydrolysis (AOAC Official Method 994.12 (2006), and the sulfur-containing amino acids (such as Met) require a pre-oxidation step (AOAC Official Method 985.28 (2006) [14].

Estimation of nutritional protein quality

The AA composition obtained by analysis allowed the nutrition value of the protein to be assessed and measured by the following indicators [16]:The Essential Amino-Acid scores (EAAS). This method involves determining an indicator of exogenous content limiting amino acid in the tested protein (a_p_) to the content of the same amino acid in a standard protein (a_s_). It was calculated according to the following Equation (1):
(1)$$EAAS=\frac{a_{p}}{a_{s}}$$

The Essential Amino-Acid Index (EAAI). It was calculated based on the following Equation (2):


(2)
$$EAAI\;\left(\%\right)=\sqrt[{n}]{\left(Lys\;1_{p}\right)/\left(Lys\;1_{s}\right)\;x\left(Thr\;2_{p}\right)/\left(Thr\;2_{s}\right)\;x\ldots x\;\left(His\;n_{p}\right)/\left(His\;n_{s}\right)}\times 100$$


As a standard for calculating the abovementioned indicators, the FAO/WHO/UNU, 2007 [17] was adopted.

Fatty acid composition

The determination of fatty acid composition was carried out by Gas Chromatography (HP 6890 II with Flame Ionization Detector) according to PN-EN ISO 5508:1996 [18] and PN-EN ISO 5509:2001 [19]. BPX 70 high-polar capillary column (60 m × 0.25 mm, 25 μm) was used for the separation of esters. Conditions of analysis: column temperature between 140 and 210 °C, doser temperature: 210 °C, detector temperature 250 °C, and helium as a carrier gas.

#### 2.2.2. Antioxidant Activity

DPPH radical scavenging assay

The antioxidant activity was determined using the modified Brand-Wiliams et al. (1995) method [20] and Alothman, Bhat and Karim, (2009) [21]. It was carried out using synthetic DPPH radical (2,2-diphenyl-1-picrylhydrazyl) (Sigma Aldrich, Darmstadt, Germany). DPPH solution prepared by dissolving 0.012 g DPPH (M = 394.32 g/mol) in 100 cm^3^ of ethanol. Extracts were prepared by pouring 25 g of crushed bar samples into 100 mL of ethanol. It was shaken for 20 h and then filtered. The solution was stored in the dark. The absorbance A0 was measured by adding up to 0.2 cm^3^ of DPPH solution to 0.8 cm^3^ ethanol. The test sample contained 0.2 cm^3^ of DPPH solution and 0.6 cm^3^ ethanol and 0.2 cm^3^ of the extract under test. After 5 and 30 min of initiation of the reaction, absorbance (A) was measured. Each measurement was performed three times and the mean absorbance value (AR) for the solution was calculated. The violet color the DPPH solution disappeared when it was reduced by test samples of bars. The decrease in absorbance was recorded by spectrophotometer GenesysTM 20 (Thermo Scientific Co., Waltham, MA, USA) at wavelength = 517 nm. The used equipment was calibrated using ethanol. The inhibition percentage of DPPH radical discoloration was calculated using the following Equation (3):% Inhibition = 100 (A0 − (-)Ar)/A0(3)
where A0 is the absorbance of the control; Ar is the absorbance of the extract.

ABTS radical scavenging assay

The ABTS (2,2′-azino-bis 3-ethylbenzothiazoline-6-sulphonic acid, Sigma Aldrich, Darmstadt, Germany) radical scavenging assay was carried out according to Re et al. (1999) [22]. After 6 min of incubation with the extracts, changes in the concentration of the ABTS•+ radical cations were assessed using a spectrophotometer, the GenesysTM 20 (Thermo Scientific Co., Waltham, MA, USA). These extracts’ antioxidative components decreased the levels of ABTS•+, which were determined by the decrease in the solution’s absorbance at 734 nm wavelength. The ability of the extracts to counteract the oxidation reaction was calculated from the formula using Equation (3).

#### 2.2.3. Statistical Analysis

The statistical analysis was performed using Statistica 13 (StatSoft Inc., Tulsa, OK, USA). A One-Way Analysis of Variance (ANOVA) was used for the statistical evaluation of the results of the physico-chemical analysis of organic bars (the significance level *p* = 0.05 was assumed). Differences among obtained means were tested by Tukey’s honestly significant difference test (Tukey’s HSD). Results are expressed as mean ± standard deviation of replicated samples.

The System Cluster Analysis was used to identify main groups of amino acids in protein bars on the ground of their size. Ward’s method based on a classical sum-of-squares criterion, producing groups that minimize within-group dispersion at each binary [23], was used. The data set consisted of a 27 × 19 matrix, in which rows represented the amino acid composition of protein bars during storage and columns represented the mean values of particular amino acid content in products.

Spearman’s rank-order correlation coefficient was calculated to characterize the relationship between the antioxidant capacity of protein organic bars detected by DPPH and ABTS assays, at the level of statistical significance *p* = 0.05, for two-sided testing.

The Principal Component Analysis was applied to interpret the fatty acids profile of HPBs stored at a controlled time (for three months) and two temperature conditions (4 °C and 22 °C). It was also applied for the summary results of changes estimation of physico-chemical quality and antioxidant activity of organic protein bars during storage. When analyzing high-dimensional data, this statistical method is frequently used to condense multivariate datasets and reveal their hidden structure [24].

## 3. Results and Discussion

### 3.1. Proximate Chemical Composition

The HPBs were developed in accordance with a previous study [13] wherein nine formulas of bars were tested in light of physicochemical, microbiological, sensory, and consumer assessment criteria. The formula used to investigate the bars in the present study was slightly modified compared with previous research in order to obtain an organic product with high nutritional value and antioxidant activity. The fresh HPBs contained in the recipe ingredients with high caloric value such as dark chocolate, dried fruits, and coconut shreds (Table 1), which had an impact on the obtained caloric values, ranging from 372 kcal per 100 g of product (B3) to 386 kcal per 100 g of product (B1), respectively (Table 2).

Such products can be considered “meal replacement” foods, used as energy sources alone or in combination with other foods. A majority of commercial meal replacement products contain vitamins and minerals and provide a nutritionally balanced, low-fat, low-energy meal plan to replace one or two regular meals or snacks per day [25].

Whey protein concentrate (WPC 80) with approximately 80 g protein per 100 g was incorporated into model organic protein bars formulated at 15.9–16.7 g protein per 100 g of products (Table 1). After consideration of protein from other ingredients included in the formulations, the total protein content in fresh bars ranged from 17.5 to 19.0 g per 100 g of product, respectively (Table 2).

The values of available carbohydrate content in fresh samples of HPBs were found in the range of 23.5–27.1 g/100 g (Table 2) and differed significantly (*p* < 0.05) among products. The highest available carbohydrate values were obtained for samples B2 and B3. It should be highlighted that only natural sources of sugars—dried fruits and honey—in the production of the bars were used, thanks to which the addition of sucrose or sweeteners was not necessary for the recipe. However, there were no significant differences between the new products in terms of reducing sugars (*p* > 0.05).

The fat content investigated in the study of fresh HPBs ranged from 17.9–21.0 g per 100 g of product, respectively (Table 2). The results showed that only the fat content in batch B1 was found to be significantly higher due to the addition of coconut shreds into this formulation.

The type of sample, time, and temperature of storage did not diversify the analyzed HPBs in terms of protein content. An individual who engages in regimented resistance training should be aware of the maximum amount of protein that can be utilized in one meal for constructing lean tissue. Muscle–protein synthesis is predicted to be maximized in young adults who consume 20–25 g of high-quality protein per day; anything above this amount may be oxidized for energy or transaminated to urea form and other organic acids [26].

Over a storage time, significant reductions in fiber content were observed in individual batches of products. However, all tested products, both fresh and stored at different temperatures, included dietary fiber above 10 g per 100 g of products (Table 2) and could therefore be considered as products with a high content of that ingredient (Regulation (EC) No 1924/2006) [27].

In contrast, the fat content of investigated HPBs did not change during storage (*p* > 0.05) (Table 2). The time of storage did not significantly affect (*p* > 0.05) the caloric value or the ash content of the various series of products. The results showed that the type of HPBs as well as temperature and storage time significantly affected (*p* < 0.05) the changes occurring in moisture content (Table 2). A significant increase (*p* < 0.05) in moisture content was only observed for the B2 series after storage for 3 months at 4 °C.

Even small changes in air humidity can contribute to changes in moisture content in food products. As reported Ekielski et al. [28], the increase in air humidity caused moisture from the air to be absorbed by the product under test, while at the same time causing changes in the mass of the product. The highest mass gain and mass growth rate were observed at 80% relative air humidity at 25 °C.

The moisture content constitutes a very important factor in determining the shelf stability of food products. The change in the moisture content observed during storage time may be the result of the nature of used packaging material, storage conditions (temperature and humidity), and hygroscopic nature of the food product [29,30].

On the other hand, Jan et al. [31] observed that nature of the used packaging material affects the selected quality parameters such as moisture content, water activity, and aspects of sensory and microbial analysis. After 60 day storage (at ambient conditions of temperature 25  ±  2  °C and a relative humidity of 68.5%) of cakes made from germinated chenopodium (*Chenopodium album*), packed into metalized polyester polyethylene (MET-PPE), a moisture increase of 4.5% was observed. However in the setting of used ingredient—flour—this increase was 7.25%.

Additionally, Banach et al. (2014) [32] reported that high-protein nutrition bars formulated with milk protein concentrate had increased moisture (ranging from 15.4 to 16.8 g/100 g) content after 42 days of storage at room temperature (~22 °C). These findings were explained by higher relative humidity at room temperature.

### 3.2. Amino Acid Composition

The changes in amino acid (AAs) composition of prepared HPBs during storage and amino acid composition of the used WPC-80 are presented in Table 3.

It was found that the amino acid profile of the WPC-80 is favorable from the point of view of the high amino acid content of exogenous amino acids: Lys, Leu, Ile, Val, and the presence of Arg and His in a smaller quantity. Data showed that 17 amino acids were identified in organic protein bars including 10 essential amino acids (EAA). The B2 bars contained most of the AAs (200.0 mg/g of protein) (Table 3).

Whey protein is one of the most frequently used proteins in food bars because of its nutritional and sensory characteristics. However, the shelf life of these functional food products is limited by quality losses during storage [33]. The main source of protein in the formulations of analyzed products was WPC-80. Therefore, the quantitative changes in the amino acid composition of organic bars may be partly due to changes in the WPC-80.

These findings are in agreement with Gasparini et al. (2020) [34]. The authors reported that the impact of the WPC storage conditions on the amino acid composition, and the nutritional value, was low. During storage of WPC, they observed losses of Ser, Tyr, and Thr amino acids. These AAs are susceptible to degradation during food processing, dehydration, and oxidative reactions [35].

The dendrogram of system cluster analysis (SCA) of amino acid composition results in analyzed HPBs is shown in Figure 1.

One of the main aspects of long-term food storage is also the protein–fatty interaction and the nutritional consequences of changes in amino acid composition, especially exogenous ones, and protein digestibility. According to Mercier et al. (2004) [36], lipid oxidation products have a destructive influence on the changes in the available forms of Lys, Met, Cys, and Tyr.

The cluster analysis was a helpful tool to analyze data on amino acid content in various products. Three groups were obtained from SCA, in which the clustering distance was 1.5. The following amino acid contents were included in the isolated groups: Group 1: Lys, Pro, Thr, Val, Arg, Ser, Ile, Ala, Met, His, Cys, Tyr, Phe, Gly (the range of content between 0.31% and 1.35%); Group 2: Leu, Asp (the range of content between 1.55 and 2.03 mg/g of protein); Group 3: Glu (the range of content between 3.32 and 4.02 mg/g of protein).

The results obtained in SCA are confirmed by the data shown in Table 3. The amino acid present in the samples in the largest amount was Glu, which was at the same time the amino acid dominant in terms of the content of used WPC-80 (Table 3). This is also due to the high amount of WPC-80, (15.9–16.7 g per 100 g of product) among all components of recipes (Table 1).

The second AA dominant in quantitative terms in the HPBs was Asp (N-EAA) (Figure 1). This AA was derived from WPC-80 and dried fruits, where it is likely to form Asp by hydrolysis by non-enzymatic brown-up [37]. Group 2 also qualified (Figure 1) Leu (EAA), of which the source was WPC–80 and the recipe components such as chocolate. The batches of products were also most closely related to Leu and Asp, which resulted from the components used in the recipes (Table 1 and Table 3).

The BCAAs are essential and perform crucial functions in the metabolism of protein and muscle components. Recent findings also suggest that using BCAA supplements could help reduce skeletal muscle atrophy in those with cardiovascular disease [38]. The BCAAs Leu, Ile, and Val were delivered mainly by WPC-80. In the examined products, Leu (1.62–1.93 mg/g of protein) appeared as the largest of all BCAAs (Table 2). Almeida et al. (2013) [39] suggested that whey protein supplements have been recognized for their high nutritional quality, fast absorption, and as a rich source of EAAs, mainly BCAAs.

When comparing the chemical indices of protein evaluation, attention should be paid to the used reference protein. The quality of investigated bars’ protein was determined by comparing it to the FAO/WHO/UNU (2007) [17]-recommended pattern of EAAs.

Limiting AA has traditionally been considered for those in the shortest supply relative to the need for protein synthesis. The first limiting AA is the one in the shortest supply relative to need. The second limiting AA is the one in the second-shortest supply relative to need, etc. [40]. The concentrations of all EAAs determined by EAA scores (AAS), except His were higher than the FAO/WHO/UNU (2007) pattern for the WPC-80. On the other hand, the protein chemical score (FAO/WHO/UNU, 2007) ranged between 0.21 and 0.48, obtained in all investigated high-protein bars, respectively (Table 2). The first limiting AA in developed products was Val, despite the fact that the differences between samples were relatively small (Table 3) and did not change during storage. It can be a result of the significant amount of the ingredient—pumpkin seeds—being used in the recipe. According to El-Soukkary (2001) [41], Val and Lys were the first limiting AAs for pumpkin protein concentrate.

Each type of protein has a limiting AA, but the kind of limiting AA differs depending on the source of protein [42]. According to Ha and Zemel (2003) [43], Lys is the first limiting AA in many high-protein foods. In products based on both dairy and soy proteins, the reduction in active Lys induced by the Maillard reaction leads to a significant quality loss during storage. On the other hand, Ammar et al. 2020 [44] reported that nutritional value determined by the limited score of the Phe + Tyr group in WPC was not significantly affected by the storage conditions.

The EAAI of WPC-80 surpassed the FAO/WHO/UNU (2007) [17] reference (Table 2). However, the EAAI values of tested products were relatively low, ranging between 30.5% and 33% in fresh products, respectively (Table 3) and during storage did not change significantly. Brown and Jeffrey (1992) [43] reported that a protein has high quality when the EAAI value is greater than 90%, moderate quality when the EAAI is between 70% and 89%, and low quality when the EAAI is less than 70%.

Our results indicated that the developed HPBs are not a source of full-value protein themselves, but they can complement a well-balanced diet in the form of snacks.

### 3.3. Fatty Acid Profile

Fat is a macronutrient and is needed for humans in relatively large amounts as a source of energy and fatty acids (FAs), a heat conserver, a component of cell walls, and a transport vehicle for absorption of fat-soluble vitamins A, D, E, and K, which serve as a way of insulating the body and as a shock absorber [45,46]. The mean proportion of particular FAs in organic protein bars depending on the storage conditions is shown in Appendix A. The Principal Component Analysis (PCA) graph of the FA profile in organic protein bars depending on the storage conditions is shown in Figure 2.

In this study, the collected results were analyzed using the sum of the first two principal components. A multivariate analysis of the data (Figure 2) showed that 52.9% of the variation between the samples was explained in the first two principal components.

The first component explained 31.2% of total variability and was mainly related to saturated fatty acids such as caproic, caprylic, capric, lauric, mirystic, and sum of saturated fatty acids (Σ SFAs). However, the second component explained 21.7% of total variability and was related to linoleic acid, α-linoleic (ALA) acid, the sum of polyunsaturated fatty acids Σ (PUFAs); oleic acid, the sum of monounsaturated acids (Σ MUFAs), and two of saturated fatty acids (SFAs), i.e., stearic and palmitic.

Samples were grouped into four clusters with their FA profiles and selected storage temperatures (4 °C and 22 °C) for three months of storage (Figure 2).

The first indicated group was the B1 series bars, regardless of storage temperature. These products are characterized by the high content of capric acid, myristic acid, and lauric acid. Sample B1 had the highest Σ SFAs. It ought to be noted that the addition of coconut shreds in the recipe of B1 bar (Table 1) significantly increased the sum of Σ SFAs (Figure 2 and Appendix A) compared with B2 and B3 bars, in which oat flakes were used instead.

The second cluster is constituted of fresh samples of B2 and B3 bars. In these bars the highest MUFAs among all investigated fresh products were reported, probably due to the highest content of oleic acid (C 18:1 n9c) (Figure 2 and Appendix A). That is also a result of the largest percentage in the recipe of pumpkin seeds (Table 1) in comparison with B1 bars.

Among the PUFAs, the content of linoleic acid (C 18:2 n6) is noteworthy. The highest amount of this acid was investigated in the B2 bars (Table 2; Appendix A), probably due to the highest percentage addition of sunflower oil (Table 1), which is a good source of this ingredient [47]. Kita et al. (2003) [48] suggested that during the storage of snacks, the composition of the FAs may change slightly and PUFAs are the most vulnerable to change. In our study, the storage temperature had an impact on linoleic acid (C 18:2 n6) and ALA amounts in investigated samples (Appendix A).

The third indicated cluster comprised products B2_4, B3_4, and B2_22. These products are characterized by the high content of ALA. However, the fourth cluster was sample B322, product series B3 after 3 months of storage at 22 °C. This product had, compared with others, a high content of behenic, heptadecanoic, and arachidic acid. On the other hand, in sample B3_22 the highest significant Σ SFAs (Appendix A) was noted.

The developed products were also characterized by their small content of TFAs (Appendix A). It involved stored bars (B2_22, B3_22) (Figure 2). The low level recorded for TFAs after the storage period of the products is the result of the lack of use in the manufacture of partly cured vegetable oils which are the source of these FAs. These compounds can be harmful when consumption is excessive. In our study, the percentage of linoleic acids in all investigated samples decreased during storage, regardless of the applied temperature (Appendix A). The percentage of palmitic acid content during storage increased only in B1 products, probably due to PUFA degradation (Appendix A). It has been suggested that the ratio of C18:2 to C16:0 is a valuable indicator of the level of PUFA deterioration [42]. The obtained results showed that this ratio values decreased in all samples during storage, except sample B3_22 (Appendix A).

The biggest reduction (from the initial) in the C 18:2/C16:0 ratio of 0.20 was observed in samples stored at 22 °C. The highest amount of C 18:2/C16:0 ratio among all investigated samples was noted in sample B3_22. This is probably due to a reduction in palmitic acid content by 5 (g/100 g) compared to the initial content of this acid in the fresh B3 sample (Appendix A). This fact suggested a faster rate of oxidative degradation in B3 samples.

FAs are obtained from various dietary sources that possess characteristic composition and consequently influence health outcomes. From this perspective, the FA composition should be assessed to determine their nutritional and/or medicinal value, especially in fatty-acid-rich foods, food supplements, or dietary foods. The ratio of PUFA to SFA is an index normally used to assess the impact of diet on cardiovascular health. It hypothesizes that all PUFAs in the diet can depress low-density lipoprotein cholesterol (LDL-C) and lower levels of serum cholesterol, whereas all SFAs contribute to high levels of serum cholesterol. Thus, the higher this ratio, the more positive the effect [49]. In our study, the series B2 and B3 of HBPs were the most favorable products in this meaning of the ratio of PUFA to SFA. It was higher than 0.42. However, the highest value of the ratio of PUFA to SFA was noted in the case of B3_22 bar (Appendix A).

The ratio of PUFA to SFA is too general and does not allow for an assessment of atherogenicity. Therefore, an atherogenicity index (IA) in the further evaluation was used. The IA indicates the relationship between the Σ SFAs and the Σ UFAs. The sSFAs (C12:0, C14:0, C16:0) promote lipid adhesion to the cells of the circulatory and immune systems. However, UFAs inhibit the accumulation of atherosclerotic plaques and influence the reduction of phospholipids, cholesterol, and esterified fatty acids, which is why they are considered anti-atherosclerotic [50]. According to this ralationship, products with lower IA may contribute to the reduction of total cholesterol and LDL cholesterol in plasma. The values of the IA index in different HPBs ranged from 0.58 (B3_22) to 1.1 (B1_4; B1_22), respectively (Appendix A). Therefore, the most favorable IA values were observed in the sample B3_22.

### 3.4. Changes in Antioxidant Activity

The changes in antioxidant capacity of HPBs, determined by methods based on organic radical scavenging, during 3 months of storage period under different temperature conditions are presented in Figure 3.

DPPH activity of fresh HPBs referred to as % DPPH inhibition ranged from 80.8% to 84.9%, respectively. The highest radical scavenging activity was recorded by B1, B3 bars of 84.4% and 84.9%, respectively (Figure 3). However, irrespective of applied temperature, after 3 months of storage in every sample, a significant increase in DPPH activity was noted.

For all samples, the ABTS assay detected significantly higher antioxidant capacity than the DPPH assay (Figure 3). Fresh bars ranged from 89.3% to 97.4% of ABTS inhibition. The highest ABTS activity (higher than 97%) (*p* < 0.05) was reported in the case of B1 and B2 bars (Figure 3). However, after 3 months of storage, irrespective of the temperature applied, a significant (*p* < 0.05) reduction of ABTS activity was noted. Particularly decreased activity (more than 56% of ABTS inhibition) occurred in B1 and B2 bars. The B3 bar, which contained oat flakes and dried apricots, kept a high activity during storage.

There are many bioactive components and antioxidants in fruit, which can inhibit the destructive effects of free radicals. Considering dried fruits have a low moisture content and a longer shelf life, their antioxidant activity and polyphenol content are expected to be high [51]. Szydłowska et al. (2020) [13] reported that ingredients of functional organic high-protein bars such as goji berries and dried cherries demonstrated high antioxidant activity referred to as % of DPPH inhibition, higher than 97%.

The dried dates also had a high DPPH free radical scavenging capacity (above 80%). The high antioxidant activity of tested bars can be a result of the dried fruits used in a recipe.

In turn, the antioxidant activity of whey proteins is attributed to their hydrophobic and aromatic amino acids which can stabilize electron-deficient radicals by donating protons [52,53]. Corrochano et al. (2018) [54] reported that whey products such as WPC showed lower antioxidant potency than plant antioxidants, although whey proteins can be added to food at high concentrations. It is worth noting, however, that this bioactivity is relatively insensitive to the method of processing and is increased by enzymatic hydrolysis.

The observed reduction of the antioxidant activity investigated in the present study HPBs during storage could be due to the transformation of the individual components of the products, including the oxidation of fats or protein decomposition.

The analysis of lipid oxidation of investigated HPBs (presented as TBARS index) was showed previously by Trząskowska et al. (2022) [55] Authors reported that the TBARS values of HPBs stored at 4 °C were lower than the rancidity period (1–2 mg MDA/kg sample) for food products. Their good physicochemical quality was confirmed. On the basis of the obtained results, it was concluded that the use of some ingredients, such as oat flakes or prunes, may limit the degree of fat oxidation during the manufacturing process and refrigerated storage.

Cross-correlation is an important element in the assessment of parameters related to the antioxidant properties of products. In the present study, the antioxidant capacity by ABTS assay was strongly negatively correlated with that of the DPPH assay (ρ = −0.8333; *p* < 0.005). The synergistic interactions between individual components of food result in elevated expression of antioxidant potency. Some studies in the literature reported that the amount of antioxidant compounds in food products depends on some factors, including the temperature, parameters of heat, processing, or solvent polarity, etc. [48,49].

Different solvents may result in different results when DPPH and ABTS assays are used to quantify antioxidant capacity [50]. The problem often overlooked in the antioxidant capacity by DPPH assay measurement is the presence of metal ions in food sample extracts. Transient metal ions (especially iron and copper) are actively involved in the formation of free radicals (Fenton and Haber–Weiss reactions). Polyphenolic compounds that chelate metal ions are masked and thus can also contribute to the fight against oxidative stress [56].

According to Floegel et al. (2011) [57], when used on a variety of plant foods containing hydrophilic, lipophilic, and highly pigmented antioxidant chemicals, the ABTS assay is superior to the DPPH assay.

Taking into account these findings, we believe that, in our study, the antioxidant activity decreased in bar samples during storage; however, it should be confirmed in a future chemical investigation, e.g., with HPLC applied.

Figure 4 shows the PCA plot for the summary results of changes estimation of selected physicochemical parameters and antioxidant activity of organic protein bars during storage.

In this instance, three principal components were found, the sum of which explains 65.23% of the variance of the variables. The analysis of the obtained results was carried out using the sum of the first two primary principal components. In the first two principal components, 54.37% of the variation between the samples was explained, according to a multivariate analysis of the data (Figure 4).

The first component explained 32.84% of the total variability and was connected to factors of proximate chemical composition such as total caloric value, crude fat, available carbohydrates, reducing sugars, crude fiber, total protein, moisture, all AAS, EAAI, and Σ SFAs. The second component explained 21.53% of the total variability and was mainly related to antioxidant activity (DPPH method, ABTS method), Σ MUFAs, Σ PUFAs, the value of ratio C18:2/C16:0, and ash (Figure 4).

It should be noted that the antioxidant activity also contributed to the observed changes in products’ quality during storage. The method used for the determination of antioxidant activity affected the results obtained, which is particularly evident in the case of higher ABTS for fresh products B1, B2, and, B3 (Figure 4).

Based on the conducted statistical analysis, the investigated samples of HPBs were grouped into four clusters with respect to the kind of product and storage conditions. The first group constituted samples of fresh bars (B1; B2; B3). This group is located at the lower of the plot. The attributes located in this area of the plot indicate that investigated samples of bars (B1, B2, B3) had quality similarities such as total protein and crude fiber content or antioxidant activity (% inhibition of ABTS) (Figure 4).

Further, three clusters isolated in PCA analysis are the groups of products stored irrespective of the applied temperature. However, the second group, constituted by products B1_4 and B1_22, are located at the top-right of the presented plot (Figure 4). The B1_4 sample was characterized by a high value of the EAAI index. In B1_22, the highest total caloric value was noted. The third group contains samples B2_4 and B2_22. These organic bars are distinguished by high total protein content. The fourth group is represented by stored bars (B3_4; B3_22) of the B3-series. These products contain a high content of moisture and available carbohydrates. It is also noteworthy that the sample B3_22 was characterized, among all investigated products, by the highest content of reducing sugars.

Based on the conducted research, it can be concluded that the storage time and applied temperature had a major impact on the reducing sugar content, fatty acid, and amino acid profiles of the investigated samples of HPBs.

## 4. Conclusions

This study demonstrated the effect of temperature and time of storage on selected physicochemical parameters and antioxidant activity of organic protein bars. The presented study is the first stage of wider research which also includes the evaluation of other quality factors. Although, microbiological, sensory, and textural quality, as well as chemical stability tests, should be performed during the comprehensive food shelf-life test, in the present study we tried to focus only on the changes in the nutritional value and antioxidant activity of the newly developed organic bars.

The snack bars produced from Polish organic raw materials have a well-balanced nutritional value and favorable fatty acid profile due to the high content of good quality WPC (15.9–16.7 g protein per 100 g of products), as well as high-quality ingredients (fruits, plants, oils). The advantage of these products is the absence of added sugar in the form of sucrose and other sweeteners, and the perceptible sweetness comes from natural organic raw materials, mainly dried fruits with high antioxidative activity.

Results generated in the study suggest that the proximate chemical composition of organic HPBs during storage changed slightly. However, significant changes in available carbohydrates and reducing sugars content were observed in products stored at room temperature (22 °C). Slight changes in the moisture content during storage only referred to B2 samples (increase of 1.2–2.2 g/100 g of product, compared with the fresh HBPs), which requires further research.

While analyzing changes in fatty acid composition during storage, there was a slight but significant reduction of Σ PUFAs, regardless of the applied temperature. Despite the determination of TFAs in stored protein bars, the level obtained was relatively low (0.1 g/100 g of product) compared with the equivalent of such products available on the market.

Overall, it was concluded that any identified changes in protein bar quality during 3 months of storage were slight. Therefore, it can be suggested that developed ready-to-eat organic protein snack bars should be stored at low temperature for 3 months due to rate any kind of storage-related changes. However, in the case of longer storage, further investigations should be conducted. Moreover, investigated products need some improvement in the aspect of nutritional quality of protein. This could be accomplished by modifying the used ingredients, such as protein isolate. We believe that the findings from this study can be used and applied to commercial production and storage studies in the development of novel organic protein products.

## Figures and Tables

**Figure 1 foods-11-03631-f001:**
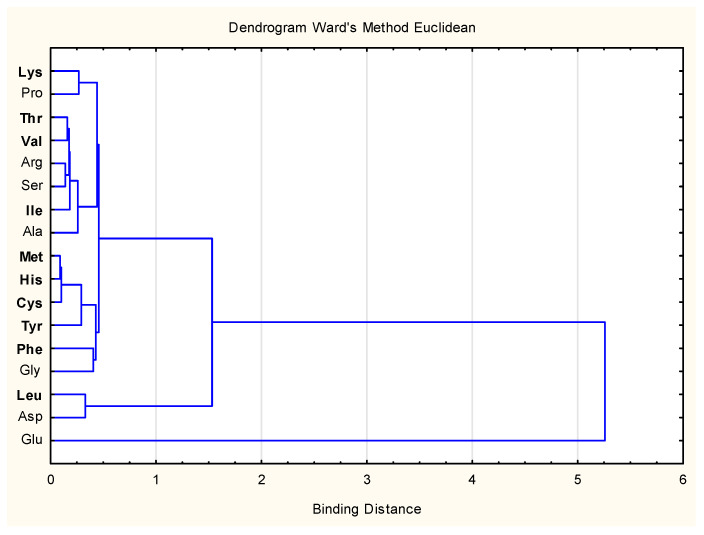
Dendrogram of system cluster analysis of amino acid composition results in analyzed organic protein bars (Ward’s method, Euclidean). Explanations: the highlighted designations of amino acids mean essential amino acids (EAA): Ala—Alanine; Arg—Arginine; Asp—Aspartic acid; Cys—Cysteine; Glu—Glutamic acid; Gly—Glycine; His—Histidine; Ile—Isoleucine; Leu—Leucine; Lys—Lysine; Met—Methionine; Phe—Phenylalanine; Pro—Proline; Ser—Serine; Thr—Threonine; Tyr—Tyrosine; Val—Valine.

**Figure 2 foods-11-03631-f002:**
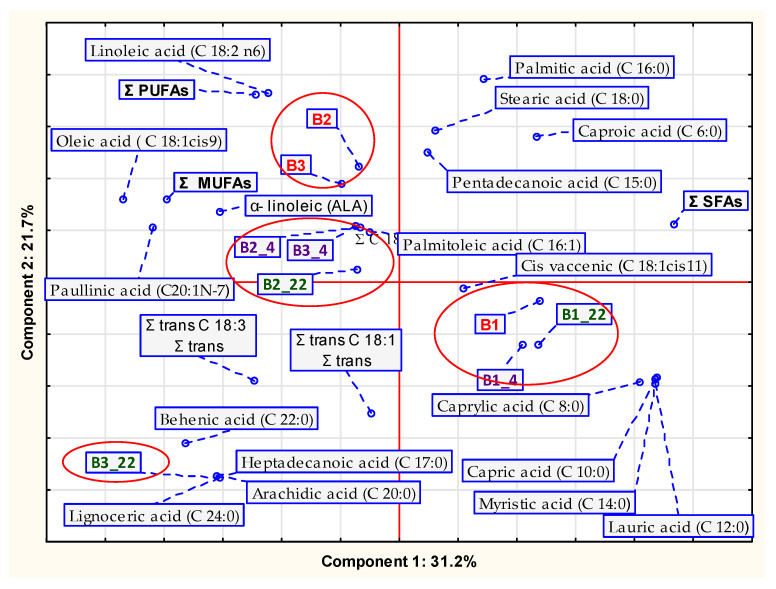
Principal Component Analysis graph of the proportion of particular fatty acids in organic protein bars depending on the storage conditions. B1; B2; B3—samples coded according to Section 2.1; Table 1; B1_4—sample B1 after 3 months storage at 4 °C, B1_22—sample B1 after 3 months storage at 22 °C, B2_4—sample B2 after 3 months storage at 4 °C, B2_22—sample B2 after 3 months storage at 22 °C, B3_4—sample B3 after 3 months storage at 4 °C, B3_22—sample B3 after 3 months storage at 22 °C.

**Figure 3 foods-11-03631-f003:**
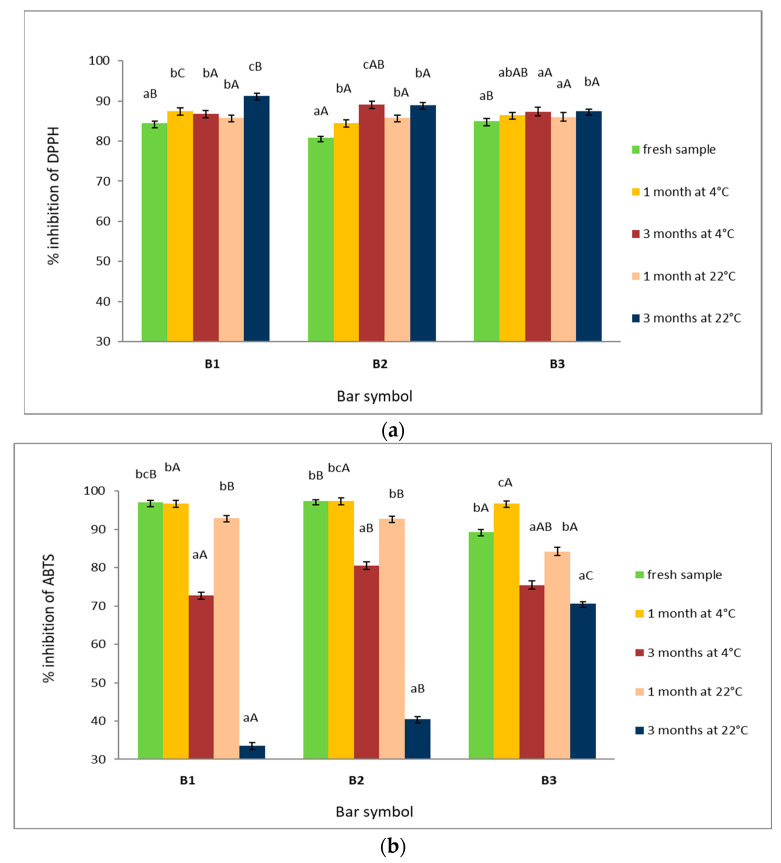
Antioxidant activity of organic protein bars stored at different temperature conditions (**a**) (% inhibition of DPPH) (**b**) (% inhibition of ABTS) during 1 week and 3 weeks. Explanatory notes: data are expressed as mean values ± standard deviations. Refer to Table 1 for identification of test samples. Values denoted by different capital letters in the same conditions of storage differ significantly (*p* < 0.05), *n* = 3. Values denoted by different lowercase letters in the same batch of bars, at different temperature storage, differ significantly (*p* < 0.05), *n* = 3. The concentration of the investigated bar in the assay cocktail (DPPH method; ABTS method—4%).

**Figure 4 foods-11-03631-f004:**
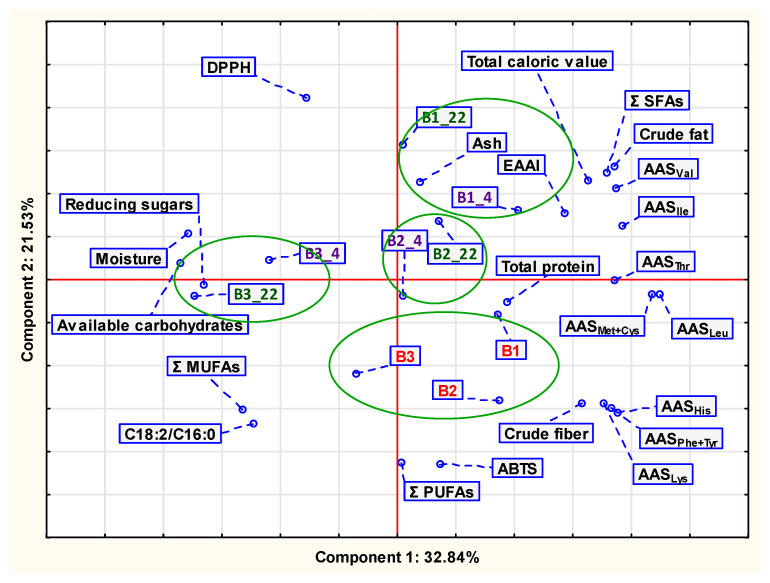
Principal Component Analysis graph of the summary results of changes estimation of physicochemical quality and antioxidant activity of organic protein bars during storage. B1; B2; B3—samples coded according to Section 2; Table 1; B1_4—sample B1 after 3 months storage at 4 °C, B1_22—sample B1 after 3 months storage at 22 °C, B2_4—sample B2 after 3 months storage at 4 °C, B2_22—sample B2 after 3 months storage at 22 °C, B3_4—sample B3 after 3 months storage at 4 °C, B3_22—sample B3 after 3 months storage at 22 °C.

**Table 1 foods-11-03631-t001:** Formulations.

Ingredients (g/100 g)	Bar Symbol		
	B1	B2	B3
Whey protein concentrate	15.9	16.7	16.3
Pumpkin seeds	14.3	15.0	14.6
Spelt flakes	11.9	12.5	12.2
Prunes	7.9	16.7	0.0
Dried apricots	7.9	0.0	16.3
Oat flakes	0.0	8.3	8.1
Coconut shreds	7.9	0.0	0.0
Honey	3.2	3.3	3.3
Sunflower oil	2.4	2.5	2.4
Inulin	2.4	2.5	2.4
Dried cherries	1.6	1.7	1.6
Freeze-dried raspberries	0.8	0.8	0.8
Water	7.9	3.3	5.7
Dark Chocolate (70% cacao)	15.9	16.7	16.3
Sum:	100	100	100

Explanatory notes: pumpkin bars (B1—pumpkin–coconut; B2—pumpkin–plum, B3—pumpkin–apricot).

**Table 2 foods-11-03631-t002:** Proximate chemical composition of organic protein bars during storage.

Fractions					Bar Symbol				
	B1	B2	B3	B1	B2	B3	B1	B2	B3
		Before storage		After 3 months at 4 °C			After 3 months at 22 °C		
Total caloric value (kcal/100 g)	386 ± 1.05 ^ABab^	384 ± 2.14 ^Aa^	372 ± 0.08 ^Aa^	383 ± 1.13 ^Ab^	382 ± 2.15 ^Ab^	375 ± 1.09 ^ABa^	389 ± 2.09 ^Bb^	384 ± 1.12 ^Bb^	377 ± 2.11 ^ABa^
* Crude fat(g/100 g)	21.0 ± 0.28 ^Ab^	19.4 ± 0.11 ^Aab^	17.9 ± 0.05 ^Aa^	21.5 ± 0.39 ^Ab^	20.0 ± 0.14 ^ABb^	17.8 ± 0.05 ^Aa^	21.4 ± 0.06 ^Ab^	19.7 ± 0.08 ^Aab^	18.2 ± 0.08 ^Aa^
* Available carbohydrates (g/100 g)	23.5 ± 0.50 ^ABa^	25.3 ± 0.17 ^Ab^	27.1 ± 0.24 ^Ab^	22.7 ± 0.31 ^Aa^	25.3 ± 0.28 ^Ab^	31.4 ± 0.13 ^Bc^	28.8 ± 0.31 ^Ba^	31.4 ± 0.25 ^Bbc^	30.6 ± 0.27 ^Bab^
Reducing sugars (g/100 g)	20.3 ± 0.33 ^Aa^	20.3 ± 0.22 ^Aa^	20.7 ± 0.35 ^Aa^	20.2 ± 0.35 ^Aa^	20.6 ± 0.31 ^Aa^	19.8 ± 0.16 ^Aa^	20.8 ± 0.34 ^Aa^	21.7 ± 0.17 ^Ba^	28.1 ± 0.37 ^Bb^
* Crude fiber (g/100 g)	16.3 ± 0.44 ^Bab^	15.4 ± 0.61 ^Ba^	15.7 ± 0.21 ^Ca^	16.0 ± 0.14 ^Bc^	13.4 ± 0.15 ^Ab^	11.1 ± 0.37 ^ABa^	10.1 ± 0.21 ^Aa^	10.8 ± 0.13 ^Aa^	10.6 ± 0.10 ^Aa^
* Total Protein (g/100 g)	17.5 ± 1.12 ^Aa^	19.0 ± 0.91 ^Aa^	17.7 ± 0.65 ^Aa^	17.5 ± 0.82 ^Aa^	18.6 ± 0.45 ^Aab^	16.9 ± 0.12 ^Aa^	18.0 ± 0.31 ^Aa^	19.4 ± 0.64 ^Aab^	17.7 ± 0.87 ^Aa^
* Ash (g/100 g)	1.8 ± 0.07 ^Aa^	1.8 ± 0.09 ^Aa^	1.8 ± 0.11 ^Aa^	1.9 ± 0.05 ^Aa^	1.8 ± 0.03 ^Aa^	1.9 ± 0.05 ^Aa^	1.8 ± 0.08 ^Aa^	1.9 ± 0.06 ^Aa^	1.8 ± 0.05 ^Aa^
* Moisture (g/100 g)	19.7 ± 0.22 ^Aa^	18.9 ± 0.21 ^Aa^	19.8 ± 0.19 ^Ba^	19.5 ± 0.12 ^Aa^	21.1 ± 0.15 ^Bb^	20.6 ± 0.11 ^Aa^	20.1 ± 0.24 ^Ab^	19.6 ± 0.19 ^Aa^	20.8 ± 0.17 ^ABbc^

Explanatory notes: * on the 100 g dry weight basis of product; Pumpkin bars (B1—pumpkin–coconut; B2—pumpkin–plum, B3—pumpkin–apricot). Values denoted by different capital letters in the same batch of bars, at different-temperature storage, differed significantly (*p* < 0.05). Values denoted by different lowercase letters in the same conditions of storage differed significantly (*p* < 0.05).

**Table 3 foods-11-03631-t003:** Amino acid composition of prepared organic protein snack bars of depending on the storage conditions.

Amino Acids	WPC 80	B1	B2	B3	Bar Symbol B1	B2	B3	B1	B2	B3	FAO/WHO/UNU Standard (2007)
**Before storage After 3 months at 4 °C After 3 months at 22 °C**											
Lys	73.8 ± 1.18	13.5 ± 0.12 ^A^	13.9 ± 0.16 ^AB^	13.5 ± 0.13 ^B^	13.1 ± 0.22 ^A^	13.5 ± 0.22 ^A^	12.0 ± 0.18 ^AB^	11.9 ± 0.19 ^A^	12.3 ± 0.20 ^A^	11.0 ± 0.18 ^A^	**45**
Met ^1^	17.5 ± 0.28	4.6 ± 0.23 ^A^	4.1 ± 0.13 ^A^	3.6 ± 0.11 ^A^	4.1 ± 0.06 ^A^	3.9 ± 0.06 ^A^	3.3 ± 0.05 ^A^	3.8 ± 0.06 ^A^	4.2 ± 0.07 ^A^	3.4 ± 0.05 ^A^	**22**
Cys ^1^	19.6 ± 0.31	4.3 ± 0.14 ^A^	3.5 ± 0.31 ^A^	3.2 ± 0.15 ^A^	4.0 ± 0.06 ^A^	3.3 ± 0.05 ^A^	3.3 ± 0.05 ^A^	3.7 ± 0.06 ^A^	3.9 ± 0.06 ^A^	3.1 ± 0.05 ^A^	
Thr	55.0 ± 0.88	9.2 ± 0.27 ^AB^	10.5 ± 0.15 ^A^	10.4 ± 0.18 ^B^	11.1 ± 0.18 ^B^	10.3 ± 0.16 ^A^	8.8 ± 0.14 ^A^	10.0 ± 0.16 ^A^	10.8 ± 0.17 ^A^	8.5 ± 0.14 ^A^	**23**
Ile (BCAA)	47.0 ± 0.75	10.0 ± 0.23 ^A^	9.3 ± 0.15 ^A^	8.8 ± 0.23 ^A^	10.2 ± 0.16 ^A^	9.5 ± 0.15 ^A^	8.2 ± 0.13 ^A^	9.1 ± 0.15 ^A^	9.8 ± 0.16 ^A^	8.3 ± 0.13 ^A^	**30**
Val (BCAA)	46.3 ± 0.74	10.3 ± 0.21 ^A^	9.60 ± 0.11 ^A^	9.40 ± 0.08 ^A^	10.9 ± 0.18 ^A^	10.2 ± 0.16 ^A^	9.0 ± 0.14 ^A^	9.70 ± 0.16 ^A^	10.60 ± 0.17 ^A^	9.0 ± 0.14 ^A^	**39**
Leu (BCAA)	85.2 ± 1.36	18.8 ± 0.30 ^A^	19.3 ± 0.12 ^AB^	16.2 ± 0.44 ^A^	18.5 ± 0.30 ^A^	18.0 ± 0.29 ^A^	15.2 ± 0.24 ^A^	17.4 ± 0.28 ^A^	18.8 ± 0.30 ^A^	15.5 ± 0.25 ^A^	**59**
His	14.4 ± 0.23	4.2 ± 0.11 ^A^	4.6 ± 0.26 ^A^	4.1 ± 0.23 ^A^	3.8 ± 0.16 ^A^	3.8 ± 0.06 ^A^	3.3 ± 0.05 ^A^	3.5 ± 0.06 ^A^	3.8 ± 0.06 ^A^	3.2 ± 0.05 ^A^	**15**
Phe ^2^	25.3 ± 0.40	7.8 ± 0.11 ^A^	8.4 ± 0.32 ^AB^	7.4 ± 0.24 ^A^	7.6 ± 0.12 ^A^	7.4 ± 0.12 ^A^	6.4 ± 0.10 ^A^	7.0 ± 0,11 ^A^	7.7 ± 0.12 ^A^	6.5 ± 0.10 ^A^	**38**
Tyr ^2^	24.5 ± 0.38	6.7 ± 0.21 ^B^	6.2 ± 0.11 ^B^	6.3 ± 0.18 ^B^	5.3 ± 0.08 ^A^	4.8 ± 0.08 ^A^	4.2 ± 0.07 ^A^	4.0 ± 0.08 ^A^	5.1 ± 0.08 ^A^	4.4 ± 0.07 ^A^	
**N-EAA (mg/g of protein)**											
Arg	21.0 ± 0.32	1.16 ± 0.21 ^AB^	1.13 ± 0.18 ^A^	0.98 ± 0.10 ^A^	1.13 ± 0.18 ^A^	1.05 ± 0.17 ^A^	0.90 ± 0.14 ^A^	1.01 ± 0.16 ^A^	1.05 ± 0.17 ^A^	0.90 ± 0.14 ^A^	
Asp	84.1 ± 1.35	1.97 ± 0.34 ^A^	1.96 ± 0.35 ^A^	1.65 ± 0.40 ^A^	2.03 ± 0.31 ^A^	1.98 ± 0.30 ^A^	1.67 ± 0.27 ^A^	1.83 ± 0,29 ^A^	1.95 ± 0.31 ^A^	1.63 ± 0.26 ^A^	
Ser	41.6 ± 0.67	1.22 ± 0.12 ^B^	1.18 ± 0.15 ^B^	0.98 ± 0.21 ^B^	1.03 ± 0.26 ^A^	1.02 ± 0.16 ^A^	0.87 ± 0.14 ^A^	1.00 ± 0.16 ^A^	1.09 ± 0.18 ^A^	0.86 ± 0.14 ^A^	
Glu	144.0 ± 2.30	4.02 ± 0.51 ^A^	3.98 ± 0.54 ^A^	3.46 ± 0.64 ^A^	3.91 ± 0,16 ^AB^	3.78 ± 0.60 ^A^	3.33 ± 0.53 ^A^	3.60 ± 0.58 ^A^	3.87 ± 0.62 ^A^	3.32 ± 0.53 ^A^	
Pro	48.1 ± 0.77	1.20 ± 0.27 ^A^	1.24 ± 0.34 ^A^	1.17 ± 0.24 ^AB^	1.25 ± 0.20 ^A^	1.17 ± 0.19 ^A^	1.05 ± 0.27 ^A^	1.22 ± 0.29 ^A^	1.27 ± 0.28 ^A^	1.07 ± 0.27 ^A^	
Gly	14.5 ± 0.23	7.1 ± 0.43 ^AB^	7.4 ± 0.27 ^AB^	6.8 ± 0.19 ^AB^	6.2 ± 0.10 ^A^	6.0 ± 0.10 ^A^	5,2 ± 0.08 ^A^	5.8 ± 0.19 ^A^	6.4 ± 0.10 ^A^	5.2 ± 0.08 ^A^	
Ala	38.1 ± 0.61	9.5 ± 0.21 ^AB^	8.3 ± 0.18 ^A^	8.5 ± 0.37 ^A^	9.0 ±0.15 ^AB^	8.6 ± 0.14 ^A^	7.4 ± 0.12 ^A^	8.2 ± 0.13 ^A^	8.9 ± 0.14 ^A^	7.3 ± 0.12 ^A^	
**Nutritional Parameters of Protein in Analyzed Treatments**											
**AAS_Lys_**	1.64	0.30	0.31	0.30	0.29	0.30	0.27	0.26	0.27	0.24	
**AAS_Met+Cys_**	1.69	0.40	0.39	0.31	0.37	0.33	0.30	0.34	0.37	0.31	
**AAS_Thr_**	1.57	0.41	0.47	0.45	0.48	0.45	0.38	0.43	0.47	0.37	
**AAS_Ile_**	1.56	0.30	0.31	0.29	0.34	0.32	0.27	0.30	0.33	0.28	
**AAS_Val_**	1.19	0.26	0.24	0.24	0.28	0.26	0.23	0.25	0.27	0.23	
**AAS_Leu_**	1.44	0.32	0.33	0.27	0.31	0.31	0.26	0.29	0.32	0.26	
**AAS_His_**	0.96	0.28	0.30	0.27	0.25	0.25	0.22	0.23	0.25	0.21	
**AAS_Phe+Tyr_**	1.31	0.38	0.38	0.36	0.34	0.32	0.28	0.29	0.34	0.29	
**EAAI (%)**	140	32.00	33.00	30.5	31.70	31.30	33.00	34.1	32.1	27.1	
**Total of EAA** **(mg/g of protein)**	408.6	89.4	89.4	82.9	88.6	84.7	69.5	81.0	87.0	72.9	**271**
**Total of N-EAA** **(mg/g of protein)**	91.4	100.7	110.6	97.7	108.7	104.6	81.2	100.6	107.6	90.3	
**Total of amino acids** **(mg/g of protein)**	500.0	190.1 ± 1.45 ^A^	200.0 ± 1.11 ^A^	180.6 ± 1.82 ^A^	197.3 ± 1.54 ^A^	189.9 ± 1.10 ^A^	172.0 ± 1.08 ^A^	181.6 ± 1.76 ^A^	194.6 ± 1.13 ^A^	163.2 ± 1.11 ^A^	**271**

Explanatory notes: Data are means ± standard deviations (*n* = 3). For each product, mean values followed by different letters within the same line differ significantly (*p* < 0.05). Refer to Table 1 for identification of test samples. The scope of accreditation of amino acid compositions 0.005–10%. Asp—the result is the sum of Asn, Asp, and its salt. Gly—the result is the sum of Gln, Gly, and its salts. Cys—the result is the sum of cystine and Cys. ^1^ Sulfur Amino Acids (Cys + Met); ^2^ Aromatic Amino Acids (Phe + Tyr). EAA: Essential amino acids; N-EAA: Non-essential amino acid; The highlighted designations of amino acids mean essential amino acids (EAA); BCAA—branched-chain amino acid. EAAI—essential amino acid index; AAS—amino acid score.

## Data Availability

Data is contained within the article or Appendix A.

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
