# Peer review of "Development of Ready-to-Eat Organic Protein Snack Bars: Assessment of Selected Changes of Physicochemical Quality Parameters and Antioxidant Activity Changes during Storage"

_foods, 2022, doi:10.3390/foods11223631_

Round 1

Reviewer 1 Report

This article is scientifically interesting in food processing. However, there are several points need to be clarified or revised. The manuscript could need a major revision and clarification of the points given as follows: 

1.   Author indicated that the overall objective of this study was to investigate the effect of temperature and a storage period on selected physicochemical parameters and antioxidant activity of organic protein bars to develop pro-healthy food for consumers. Antioxidant activity of HPB with high calorie (372 to 386 kcal /100 g) was evaluated using in vitro method such as DPPH and ABTS radical scavenging assay, which may be deficiency in corroborating that was pro-healthy food. I suggest that author should revise the description about this concept.

2.    HPB was indicated to exhibit antioxidant activity due to dried fruit used in recipe. Author should evaluate the following suggestions:

(1) Change of bioactive components with antioxidant activity should be evaluated in HPB during storage.

(2) Change of antioxidant activity is related to lipid oxidation which should be conducted and discussed in this study.

3.   The storage test of HPB in this study is 3 months and change in protein bar is slight. Longer storage test is necessary to comprehend the timing of quality deterioration.

4.  Water activity, pH, and chemical and microorganism composition may affect the changes in food products quality during storage. This study focused on the change in chemical and nutritional composition. Water activity and microorganism is important quality index, which should be conducted in this study.

Author Response

Response to Reviewer 1:

We would like to thank the Reviewer for careful and thorough reading of this manuscript and for the thoughtful comments and constructive suggestions, which help to improve the quality of this manuscript. All the changes in manuscript were marked in colors.

Comment 1: . Author indicated that the overall objective of this study was to investigate the effect of temperature and a storage period on selected physicochemical parameters and antioxidant activity of organic protein bars to develop pro-healthy food for consumers. Antioxidant activity of HPB with high calorie (372 to 386 kcal /100 g) was evaluated using in vitro method such as DPPH and ABTS radical scavenging assay, which may be deficiency in corroborating that was pro-healthy food. I suggest that author should revise the description about this concept:

HPB was indicated to exhibit antioxidant activity due to dried fruit used in recipe. Author should evaluate the following suggestions

  • Change of bioactive components with antioxidant activity should be evaluated in HPB during storage.

Response: changes of bioactive components’ content such as vitamins, flavonoids (etc.) in investigated products wasn’t estimated during storage. However, the antioxidant activity and other nutrients affecting the nutritional value such as reducing sugars, available carbohydrates, total protein(etc.) were determined. 

Comment 2: .(2) Change of antioxidant activity is related to lipid oxidation which should be conducted and discussed in this study.

Response Thank you for this suggestion. The analysis of lipid oxidation of investigated HBPs couldn’t submit in this article, because it  was presented in the previous article:

Trząskowska, M.; Neffe-Skocińska, K.; Okoń, A.; Zielińska, D.; Szydłowska, A.; Łepecka, A.; Kołożyn-Krajewska, D. Safety Assessment of Organic High-Protein Bars during Storage at Ambient and Refrigerated Temperatures. Appl. Sci. 2022, 12, 8454. https://doi.org/10.3390/app12178454

Corrections have been made In the text of article (Lines: 646-652) and  a comment regarding a changes in antioxidant activity and lipids oxidation has been added with the addition of citation of said article.  

  1. The storage test of HPB in this study is 3 months and change in protein bar is slight. Longer storage test is necessary to comprehend the timing of quality deterioration.

Response:   We thank you for this suggestion. Said matter was adressed in the section Conclusions (Lines: 747-748).

  1. Water activity, pH, and chemical and microorganism composition may affect the changes in food products quality during storage. This study focused on the change in chemical and nutritional composition. Water activity and microorganism is important quality index, which should be conducted in this study.

Response: Thank you for this comment. Of course the quality in the microbiological aspect is very important quality index. The results shown in the manuscript are a part of a larger project. Due to the fact that a huge number of results were obtained, the material was divided into two parts. The first one was published and cited in the Revised version of our manuscript and in the reply to comment 2.  Such information was presented in the different article, which was focused on estimation of safety parameters of investigated products during refrigerated storage.

Trząskowska, M.; Neffe-Skocińska, K.; Okoń, A.; Zielińska, D.; Szydłowska, A.; Łepecka, A.; Kołożyn-Krajewska, D. Safety Assessment of Organic High-Protein Bars during Storage at Ambient and Refrigerated Temperatures. Appl. Sci. 2022, 12, 8454. https://doi.org/10.3390/app12178454

Reviewer 2 Report

The work proposes the development of a high-protein snack bar.

In terms of form the document is well structured and easy to read, in relation to its contribution to the discipline I consider it discreet as it represents only a product development.

Here are some observations for improvement

It is necessary to revise the English wording and the use of some words.

L22 What does crude lipids refer to?

L50 Revise the wording.

L66 Instead of originating from foods, it should be originating from the raw materials used.

L69 The comparison with wine and cheese is not necessary.

L73-74 the sentence is obvious, check relevance.

L76 instead of using the term invest, it could be said that the food industry develops new ones .....

L83 review the concept of pro healthy foods?

L98 The title of the table is somewhat confusing, it could simply be replaced by formulations.

In table 1 it says prune, it should say prunes.

L113 The type of chocolate and its percentage of cocoa is not specified.

L125-126 The ambient humidity conditions are not specified, nor the specifications of the plastic in which the samples were packaged. Literature on the development of shelf-life tests should be reviewed. As described it is incomplete.

In the AA analyses, the equipment and configuration used is not adequately described.

Nowhere is it mentioned if there was any repetition of the measurements, only statistical analysis is mentioned.

Review the relevance/support of L236-L244, idem L270-L274.

The discussion could be improved in terms of depth, the vast majority of the discussion is limited to presenting and describing the results rather than a more in-depth analysis of the results.

The conclusions can be improved and should be related to the findings of the results.

Author Response

Response to Reviewer 2:

We would like to thank the Reviewer for careful and thorough reading of this manuscript and for the thoughtful comments and constructive suggestions, which help to improve the quality of this manuscript. All the changes in manuscript were marked in colors.

Comment 1: 

Dear authors,

the manuscript presents an interesting topic about protein snack bars. I recommend the manuscript for a minor revision.

 Material and Methods

L123: add more details of packed

Response: thank you for this suggestion. The correction has been made (Subsection 2.1.2.;  line:125-128).

Comment 2:  Results and Discussions:

L308-317: Are you sure that the higher moisture in sample B2 storage for 3 months at 4 °C was based on water vapor in the air condensing? In my opinion, if it is true, this phenomenon would also be present in samples B1, and B3 storage 3M at 4 °C. It is speculation. Also, in the next paragraph, you cited a research paper where authors reported higher moisture of bar storage at room temperature which is a contradiction with L308-317. At the end of the paragraph, you wrote “…a positive correlation between moisture content and storage time observed …in non-hermetic conditions”. So please, discuss it again and think about your packaging conditions.

Response: thank you for this comment. We agree with it. The sample differed from the others in terms of composition, in this variant, an ingredient such as dried apricots wasn’t applied. However, it does not seem to be the reason for these changes. After having considered the reviewer's comment, the authors have abandoned those irrelevant reflections. It could  be a measurement error. This was not a good line of reasoning.

 A correction regarding conditions of packaging has been made (Subsection 2.1.2.;  line:125-128). Also a new part of discuss (lines: 316-332) was added.

Comment 3:  It will be good to add the results of the selected parameters determined in more ingredients, not only WPC-80. Especially, when you discuss e. g. AA in pumpkin seeds or FAME in coconut shreds.

Response:

Thank you for this suggestion. However, the scope of the presented study included the evaluation of finished products, not raw materials. In addition, the evaluation of amino acid composition was carried out for the whey protein concentrate (WPC 80) because it was a used functional ingredient  with the highest protein content. (approximately 80 g protein according to manufacturer declaration).

Comment 4: L496: correct: “capric acid, capric acid”

Response: The correction has been done.

Comment 5:

Conclusion:

L709-711 it contradicts the abstract L32-33.

  • Response: We sincerely thank you for this suggestion. I am sorry for the mistake. The correction has been made in the section Conclusions (lines: 746-747).

Reviewer 3 Report

Dear authors,

the manuscript presents an interesting topic about protein snack bars. I recommend the manuscript for a minor revision.

 Material and Methods

L123: add more details of packed

 Results and Discussions:

L308-317: Are you sure that the higher moisture in sample B2 storage for 3 months at 4 °C was based on water vapor in the air condensing? In my opinion, if it is true, this phenomenon would also be present in samples B1, and B3 storage 3M at 4 °C. It is speculation. Also, in the next paragraph, you cited a research paper where authors reported higher moisture of bar storage at room temperature which is a contradiction with L308-317. At the end of the paragraph, you wrote “…a positive correlation between moisture content and storage time observed …in non-hermetic conditions”. So please, discuss it again and think about your packaging conditions.

It will be good to add the results of the selected parameters determined in more ingredients, not only WPC-80. Especially, when you discuss e. g. AA in pumpkin seeds or FAME in coconut shreds.

L496: correct: “capric acid, capric acid”

Conclusion:

L709-711 it contradicts the abstract L32-33.

Author Response

Response to Reviewer 3:

We would like to thank the Reviewer for careful and thorough reading of this manuscript and for the thoughtful comments and constructive suggestions, which help to improve the quality of this manuscript.  All the changes in manuscript were marked in colors.

The work proposes the development of a high-protein snack bar. In terms of form the document is well structured and easy to read, in relation to its contribution to the discipline I consider it discreet as it represents only a product development.

Here are some observations for improvement:

Comment 1: It is necessary to revise the English wording and the use of some words.

Response:

Thank you for this suggestion. The corrections have  been done.

Comment 2: L22 What does crude lipids refer to?

Response: The „crude lipids” refers to content of lipids. The word “crude” refers to the a method for the determination of this nutrient.

Comment 3: L50 Revise the wording.

Response: The correction has been done.

Comment 4: L66 Instead of originating from foods, it should be originating from the raw materials used.

Response: The correction has been done.

Comment 5: L69 The comparison with wine and cheese is not necessary.

Response: Thank you for this suggestion. This part of the text was deleted.

Comment 6: L73-74 the sentence is obvious, check relevance.

Response: The relevance was checked.  A suitable correction has been done.

Comment 7: L76 instead of using the term invest, it could be said that the food industry develops new ones .....

Response: the correction has been done.

Comment 8: L83 review the concept of pro healthy foods?

Response: Pro healthy food (functional food) refers to dietary items that, besides providing nutrients and energy, beneficially modulate one or more targeted functions in the body, by enhancing a certain physiological response and/or by reducing the risk of disease.

Comment 9: L98 The title of the table is somewhat confusing, it could simply be replaced by formulations.

Response: Thank you for this suggestion. The correction has been done.

Comment 10: In table 1 it says prune, it should say prunes.

Response: the correction has been done (Table1).

Comment 11: L113 The type of chocolate and its percentage of cocoa is not specified.

Response: Thank you for this comment. It was dark chocolate with 70% of cacao content. This information was completed in Table 1.

Comment 12: L125-126 The ambient humidity conditions are not specified, nor the specifications of the plastic in which the samples were packaged. Literature on the development of shelf-life tests should be reviewed. As described it is incomplete.

Response: in this study an ambient humidity conditions weren’t estimated. However, a correction regarding  specifications of the plastic in which the samples were packaged, was completed (Subsection 2.1.2. lines:125-128). The literature on the development of shelf-life tests  was also reviewed (Lines: 316-332).

Comment 13: In the AA analyses, the equipment and configuration used is not adequately described.

Response: the correction has been done (lines: 152-154).

Comment 14: Nowhere is it mentioned if there was any repetition of the measurements, only statistical analysis is mentioned.

Response: This information is presented in subsection 2.1 Materials, lines: 96 - 97. All analyses were performed in three independent replicates.

Comment 15: Review the relevance/support of L236-L244, idem L270-L274.

Response:  The relevance support was reviewed (line 275).

Comment 16: The discussion could be improved in terms of depth, the vast majority of the discussion is limited to presenting and describing the results rather than a more in-depth analysis of the results.

Response: We sincerely thank you for this suggestion. Since this Article has a Section 3 entitled "Results and Discussion", the discussion of results was completed (lines: 316-332; 646-652). The form of discussion adopted by the authors was also an analysis of the presented diagrams, which summarize the obtained results: Figure 1; Figure 2 and Figure 3.

Comment 17: The conclusions can be improved and should be related to the findings of the results.

Response: Thank you for this comment. The  section “Conclusions”  was improved according to the reviewer’s comment (lines: 728; 730; 736-738; 742).

Round 2

Reviewer 1 Report

The organic protein bars with antioxidant activity may be pro-healthy food for consumers described in Line 81-85. But, it exhibited high calorie (372 to 386 kcal /100 g). I suggest that author should emphasize antioxidant activity on the changes during storage. Could author revise the description about this concept?

Author Response

Response to Reviewer 1:

Comment 1:

The organic protein bars with antioxidant activity may be pro-healthy food for consumers described in Line 81-85. But, it exhibited high calorie (372 to 386 kcal /100 g). I suggest that author should emphasize antioxidant activity on the changes during storage. Could author revise the description about this concept?

Response: Thank you for this suggestion.

The investigated HPBs are an example of high protein food,  one  kind of the functional food. In the investigated  bars, the caloric value was ranged from 372 to 386 kcal/100 g of product , but it  did not deviate from the caloric value recorded in such products available on the market (275 to 438 kcal/100 g of product) [1]. The aim of presented study wasn’t to develop a final product with reduced caloric value. The HPBs can be consumed by people with high physical, sport activity for supporting health. However, such products can suitably replace consumed energy and nutrients immediately after physical engagement while saving meal preparation time.

The changes in antioxidant activity  during HPBs’ bars were described in more details  in the text of the manuscript (lines: 612-624).

In addition, the antioxidant activity determination has contributed to changes in the quality of the products evaluated during storage. A corresponding comment has been added to the text (lines: 698-701; 707).

  1. Jovanov P, Sakač M, Jurdana M, Pražnikar ZJ, Kenig S, Hadnađev M, Jakus T, Petelin A, Škrobot D, Marić A. High-Protein Bar as a Meal Replacement in Elite Sports Nutrition: A Pilot Study. Foods. 2021 Oct 29;10(11):2628. doi: 10.3390/foods10112628. PMID: 34828911; PMCID: PMC8617883.

Reviewer 2 Report

The authors have provided satisfactory responses to the reviewers' comments.

Author Response

Response to Reviewer 2

The authors have provided satisfactory responses to the reviewers' comments.

Response: We would like to express our gratitude to the Reviewer for reading this article carefully and thoroughly as well as for providing meaningful feedback and helpful suggestions that helped to elevate the article's quality.